# Epigenetic suppression of SLFN11 in germinal center B-cells during B-cell development

**Fumiya Moribe[1], Momoko Nishikori[1]\*, Tsuyoshi Takashima[2], Daiki Taniyama[3], Nobuyuki Onishi[4], Hiroshi Arima[1], Hiroyuki Sasanuma[5], Remi Akagawa[5], Fathi Elloumi[6], Shunichi Takeda[5], Yves Pommier[6], Eiichi Morii[2], Akifumi Takaori-Kondo[1], Junko Murai[7]\***

**1** Department of Hematology and Oncology, Graduate School of Medicine, Kyoto University, Kyoto, Japan, **2** Department of Pathology, Graduate School of Medicine, Osaka University, Osaka, Japan, **3** Department of Molecular Pathology, Graduate School of Biomedical and Health Sciences, Hiroshima University, Hiroshima, Japan, **4** Division of Gene Regulation, Institute for Advanced Medical Research, Keio University School of Medicine, Tokyo, Japan, **5** Department of Radiation Genetics, Graduate School of Medicine, Kyoto University, Kyoto, Japan, **6** Developmental Therapeutics Branch and Laboratory of Molecular Pharmacology, Center for Cancer Research, National Cancer Institute, NIH, Bethesda, MD, United States of America, **7** Institute for Advanced Biosciences, Keio University, Tsuruoka, Yamagata, Japan

\* muraij@ttck.keio.ac.jp (JM); nishikor@kuhp.kyoto-u.ac.jp (MN)

## Abstract

### Background

SLFN11 has recently been reported to execute cancer cells harboring replicative stress induced by DNA damaging agents. However, the roles of SLFN11 under physiological conditions remain poorly understood. Germinal center B-cells (GCBs) undergo somatic hypermutations and class-switch recombination, which can cause physiological genotoxic stress. Hence, we tested whether SLFN11 expression needs to be suppressed in GCBs during B-cell development.

### Objective

To clarify the expression profile of SLFN11 in different developmental stages of B-cells and B-cell-derived cancers.

### Methods

We analyzed the expression of *SLFN11* by mining cell line databases for different stages of normal B-cells and various types of B-cell-derived cancer cell lines. We performed dual immunohistochemical staining for SLFN11 and B-cell specific markers in normal human lymphatic tissues. We tested the effects of two epigenetic modifiers, an EZH2 inhibitor, taze-metostat (EPZ6438) and a histone deacetylase inhibitor, panobinostat (LBH589) on SLFN11 expression in GCB-derived lymphoma cell lines. We also examined the therapeutic efficacy of these drugs in combination with cytosine arabinoside and the effects of SLFN11 on the efficacy of cytosine arabinoside in SLFN11-overexpressing cells.

### Results

*SLFN11* mRNA level was found low in both normal GCBs and GCB-DLBCL (GCB like-dif-fuse large B-cell lymphoma). Immunohistochemical staining showed low SLFN11

**Data Availability Statement:** All the data for the figures and supporting information including unprocessed blot images in this manuscript are

available from the Mendeley database (https://data.mendeley.com/datasets/kn726mznn3/4).

**Funding:** This work was supported by Grants-in-Aid for Scientific Research (JP18K08324 to M.N. and JP19H03505 to J.M.), a research grant from The Uehara Memorial Foundation (to J.M.), a Grant-in-Aid from the Ministry of Education, Science, Sport and Culture (KAKENHI 19K22561 and 16H06306 [to S.T.] and KAKENHI 18H04900 and 19H04267 [to H.S.]), the Japan Society for the Promotion of Science Core-to-Core Program, Advanced Research Networks (to S.T.) and the Center for Cancer Research, the Intramural Program of the National Cancer Institute (Z01-BC-006150 to Y.P.) and research funds from the Yamagata prefectural government and the City of Tsuruoka (to J.M.).

**Competing interests:** We have read the journal's policy and the authors of this manuscript have the following competing interests: M.N. and A.T-K received honorarium and research funding from Eisai Co., Ltd. Other authors declare no conflicts of interest. This does not alter our adherence to PLOS ONE policies on sharing data and materials.

expression in GCBs and high SLFN11 expression in plasmablasts and plasmacytes. The EZH2 and HDAC epigenetic modifiers upregulated SLFN11 expression in GCB-derived lymphoma cells and made them more susceptible to cytosine arabinoside. SLFN11 overexpression further sensitized GCB-derived lymphoma cells to cytosine arabinoside.

## Conclusions

The expression of SLFN11 is epigenetically suppressed in normal GCBs and GCB-derived lymphomas. GCB-derived lymphomas with low SLFN11 expression can be treated by the combination of epigenetic modifiers and cytosine arabinoside.

## Introduction

Members of the *Schlafen* (*Slfn*) family are specific to mammals. *Schlafen* family members in mice (*Slfn1*, *2, 3, 4, 5, 8, 9, 10 and 14*) only partially overlap with those in humans (*SLFN5*, *11*, *12*, *13 and 14*) [1]. While mouse *Slfns* have been reported to function in immune response and lymphocyte development, expression and function of human *SLFNs* in lymphocytes have not been studied [2–4].

SLFN11, a putative DNA/RNA helicase, has recently been analyzed for its functions in DNA damage response [5, 6], restriction factor against replication stress [7, 8], RNA cleavage activity [9], and defense against viral infection [10–12]. As for its role in DNA damage response, independent studies have shown that SLFN11 augments the sensitivity of cancer cells to a wide range of DNA-damaging agents including platinum-derivatives, topoisomerase inhibitors, PARP inhibitors and replication inhibitors [4–6, 13–16]. Clinical studies indicate the potential value of SLFN11 as a predictive biomarker for the response to these drugs in lung and breast cancers [17, 18]. Mechanistically, SLFN11 binds to chromatin at stressed replication forks that are generated after DNA damage where it selectively blocks fork progression, and consequently induces cell death [7]. Hence, SLFN11 has come forward for its significant role as executor of cells harboring genotoxic stress.

B-cells undergo gene editing at variable regions of the immunoglobulin gene loci during the development and maturation. During this process, B-cells are physiologically exposed to genotoxic stress caused by somatic hypermutations and class-switch recombination [19, 20]. Such genotoxic stress is introduced particularly to centroblasts and centrocytes in germinal centers (GCs) of lymph nodes by activation-induced cytidine deaminase (AID) to generate different antibodies [21]. AID further induces DNA deamination at non-targeted genes [22]. Thus, germinal center B-cells (GCBs: centroblasts and centrocytes) are presumably exposed to genotoxic stress. Here, we hypothesized that the expression of SLFN11 needs to be controlled during B-cell development to avoid SLFN11-dependent cell death in cells undergoing genomic rearrangements.

The expression level of SLFN11 widely varies among cell types and tissues [6, 23]. Its expression has been shown to be largely regulated by epigenetic modifications of DNA and/or histones at its promoters, whereas gene copy number alterations and deleterious mutations of *SLFN11* have been rarely reported [24–26]. Hence, SLFN11 expression can be activated by epigenetic modifiers such as inhibitors for DNA methyltransferases, EZH2 a histone methyltransferase, and histone deacetylases [24–26].

The aim of this study was to clarify the expression pattern of SLFN11 in B-cells at different stages of development and differentiation and the potential roles played by SLFN11 in B-cells. We show that SLFN11 expression is epigenetically driven during B-cell development, and is typically suppressed in GCBs. Moreover, we show that epigenetic activation of SLFN11 in

lymphomas of GCB origin enhances their susceptibility to the clinical DNA-damaging agent cytosine arabinoside, which targets DNA replication.

## Materials and methods

### Analysis of gene expression data sets

Microarray gene expression data derived from flow-sorted B-cell subsets in human bone marrow and tonsil were obtained from NCBI's Gene Expression Omnibus database (GSE68878 and GSE69033) [27]. The exon array data were RMA normalized using R/BioC and a custom Chip Description File (CDF) [28, 29].

RNA-sequence gene expression data derived from 1001 diffuse large B-cell lymphoma (DLBCL) samples and the core set of 624 DLBCL samples was obtained from EGA (dataset id: EGA00001003600). Gene expression was measured using terms of fragments per kilobase of exon per million fragments mapped and normalized using the Cufflinks package, version 2.2.1 [30]. Quantile normalization was performed, and the data were log2 normalized.

Gene expression data, RNA-seq data, drug activity data were obtained from Genomics of Drug Sensitivity in Cancer (GDSC: https://www.cancerrxgene.org) and the Cancer Cell Line Encyclopedia (CCLE: https://portals.broadinstitute.org/ccle) using CellMinerCDB (https://discover.nci.nih.gov/cellminercdb/) [31].

### Human tissue samples

For immunohistochemical (IHC) staining analysis, we used formalin-fixed paraffin-embedded (FFPE) lymphatic tissue samples from eight cancer patients. All samples were obtained from the archives of the National Hospital Organization Kure Medical Center and Chugoku Cancer Center with the informed consent for the patients. This study was approved by the Ethics Committee of Kure Medical Center and Chugoku Cancer Center, Kure, Japan (No. 2019–36) and conformed to the ethical guidelines of the Declaration of Helsinki.

### Immunohistochemical staining

The antibodies used for IHC were as follows; mouse monoclonal anti-SLFN11 antibody (D-2, #sc-515071, Santa Cruz, 1:50 dilution), mouse monoclonal anti-CD3 antibody (F7.2.38, #20019562, DAKO, 1:400 dilution), mouse monoclonal anti-CD20cy antibody (L26, #00083951, DAKO, 1:800 dilution), mouse monoclonal anti-CD138 antibody (MI15, #00046047, DAKO, 1:100 dilution) and rabbit monoclonal anti-CD38 (EPR4106, ab108403, Abcam, 1:1000 dilution).

Formalin-fixed paraffin-embedded tissue sections (4 μm) were deparaffinized with fresh xylene for 5 min 4 times and were rehydrated with 100% ethanol, 90% ethanol, and 80% ethanol for 5 min each. Antigen retrieval was performed by pressure cooker in EnVision FLEX TARGET RETRIEVAL SOLUTION HIGH PH (50×) (DAKO) for 5 min. Endogenous peroxidase activity was blocked by incubating the sections for 10 min. For SLFN11 staining, the sections were incubated with mouse monoclonal anti-SLFN11 antibody (1:50 dilution in REAL Antibody Diluent (DAKO)) at room temperature for one hour. The sections were incubated with the second antibody (REAL EnVision Detection Reagent Peroxidase Mouse, 100 μL) at room temperature for 60 hours. The sections were incubated with DAB ENHANCER (DAKO) for 10 min. For CD markers, after incubation with the mouse monoclonal antibodies, the sections were incubated with Affinity Pure Rabbit Anti-Mouse IgG (H+L) at room temperature for one hour. The sections were treated with Stayright Purple HRP Staining substrate (#45900, AAT Bioquest) at room temperature for 10 min. The sections were counterstained

with Mayer's hematoxylin for 1 min. For scoring SLFN11-positive population (%), we manually counted the number of SLFN11-positive cells in each CD marker-positive cells for more than 100 cells and from 3 different samples.

## Cell culture and chemical compounds

The following cell lines used in this experiment were described previously [32–34]; a germinal center B-cell-like (GCB)-DLBCL line SU-DHL6; a Burkitt lymphoma (BL) line Daudi; follicular lymphoma (FL) lines FL18, FL218 and FL318. These cell lines were tested negative for mycoplasma (TaKaRa PCR Mycoplasma Detection Set; 6601), maintained in RPMI1640 (Nacalai Tesque, Kyoto, Japan) supplemented with 10% fetal bovine serum and 1% penicillin/streptomycin/L-glutamine and cultured at 37˚C in a humidified incubator in the presence of 5% CO2. An EZH2 inhibitor tazemetostat (EPZ6438) was purchased from Apexbio (Boston, MA, USA). A histone deacetylase (HDAC) inhibitor panobinostat (LBH589) was purchased from Selleck Chemicals (Houston, TX, USA). The cells were seeded at $0.2–1.0 \times 10^6$ cells in 2 ml of medium per well in 12 well plates and treated with 5 μM tazemetostat for 4 days or 10 nM panobinostat for 16 hours before being collected for RNA extraction and western blotting. Dimethyl sulfoxide (DMSO) was used as a vehicle control.

## Reverse transcription (RT)-polymerase chain reaction (PCR)

Total RNA was extracted using RNeasy Mini kit (Qiagen, Hilden, Germany) and complementary DNA (cDNA) was synthesized using SuperScript III First-Star and Synthesis system (Life Technologies, Carlsbad, CA, USA). Quantitative RT-PCR was performed using TB Green Premix Ex Taq II (Takara). Relative gene expression was normalized to ACTB expression. The sequence information of primers used for RT-PCR is available in S1 Table.

## Western blotting and antibodies

To prepare whole cell lysates, cells were lysed with RIPA lysis buffer system (Santacruz Biotechnology, TX, USA). Samples were mixed with tris-glycine SDS sample buffer (Nacalai Tesque) and loaded onto tris-glycine gels (BioRad). Blotted membranes were blocked with 4% bovine serum albumin (BSA) (Sigma-Aldrich. A9418) in phosphate-buffered saline (PBS) with 0.1% tween-20 (PBST). The primary antibody was diluted in 5% BSA/PBST by 1:3000 for SLFN11, and 1:10000 for Actin and acetyl-histone H3 (Lys9) (H3K9ac). The HRP-conjugated secondary antibody for mouse or rabbit (Cell Signaling, 7074S for rabbit and 7076S for mouse) was diluted in 4% BSA/PBST by 1:10000. After the membrane was soaked in ECL solution (BioRad), the blot signal was detected with luminescent image analyzer (LAS4000, GE healthcare). The mouse monoclonal anti-SLFN11 antibody (sc-515071X, 2 mg/ml, mouse monoclonal IgG against amino acids 154–203 mapping within an internal region of SLFN11 of human origin, Santacruz), the rabbit monoclonal anti-Actin antibody (12748S, rabbit monoclonal antibody to a synthetic peptide corresponding to residues near the carboxy terminus of human β-actin protein, Cell Signaling) and the rabbit monoclonal anti-acetyl-histone H3 (Lys9) (9649S, rabbit monoclonal antibody to a synthetic peptide corresponding to the amino terminus of histone H3 in which Lys9 is acetylated, Cell Signaling) were used.

## Cell viability experiments

Cell viability experiments by flow cytometry in Fig 5B and 5C were performed as follows: $5 \times 10^4$ cells/mL viable cells were pretreated with 100 or 500 nM tazemetostat for 4 days, and $2–5 \times 10^5$ cells/mL viable cells were pretreated with 5 or 10 nM panobinostat for 16 hours;

2–32 µM AraC was added to the cells for additional 24-hour incubation followed by evaluation of cell viability using Flow cytometry. Flow cytometry was performed using FACSlyrics (BD Biosciences, San Jose, CA, USA). Propidium Iodide Solution (Biolegend; 421301) was used for the evaluation of cell viability. Data were analyzed using FlowJo software (version 10.1; Tree Star Inc, San Carlos, CA, USA). Viability (%) of treated cells was defined as treated cells/untreated cells × 100. Combination index (CI) values were assessed using the CompuSyn Software (Combosyn Inc., Paramus, NJ) [35, 36].

For viability assay in Fig 5E, ten thousand SU-DHL6 cells were seeded in 96-well white plates (Perkin Elmer Life Sciences, 6007680) in 100 µL of medium per well. Cellular viability was determined using the ATPlite 1-step kits (PerkinElmer). Luminescence was measured by TECAN infinite M200. The ATP level in untreated cells was defined as 100%. Viability (%) of treated cells was defined as ATP level of treated cells/ATP level of untreated cells × 100.

## Plasmid construction

The doxycycline-inducible expression vector (pPCTetOn) (S1 Fig) was first made by insertion of Xho I-digested 5'- and 3'-terminal inverted repeats (IRs) cassette sequences of the piggyBac system [37] amplified by PCR using two oligos:

5'-CCGCTCGAGTTAACCCTAGAAAGATAATCATATTGTGACGTACGTTAAAGATAATCAT GCGTAAAATTGACGCATGTTCGAAATGCATGG and 5'-CCGCTCGAGTTAACCCTAGAAAGAT AGTCTGCGTAAAATTGACGCATGCGAATTCGGTACCATGCATTTCGAACATGCG into the Sal I-digested pcDNA3 vector [38] to create the pcDNA-IRs. The CAG promoter digested with Mph1103 I and Acc65 I from pCAG-BSD vector (WAKO) was cloned into pcDNA-IRs with Mph1103 I and Acc65 I to create pPC. The 3xFL-IRES-PuroR-HSV TK poly (A) signal fragment was PCR amplified from pMX-3xFL-IP [39] using primers 5'-
CGGAATTCATGGGCGTTGCCATGCCAGGTGCCGAAGATGATGTGGTGTAACAATTCATGG ACTACAAAGACCATGACGG and 5'-
ACATGCATGCGAACAAACGACCCAACACCGTGCGTTTTATTCTGTCTTTTTATTGCCGG TCGACTCAGGCACCGGGCTTGCGGG. The PCR product was cloned into pPC with EcoR I and Pae I to create pPC-IP. The Trans Activation Responsive region (TAR)- TransAcTivator (Tat) cassette was PCR amplified from pHEK293 Ultra Expression Vector (TaKaRa). TAR was amplified using primers 5'-CCTCACTAAAGGTGTACAGTACTTCAAGAACTGCTGATATC and 5'-
CTAGGATCTACTGGCTCCATGAGGCTTAAGCAGTGGGTTC, and Tat-P2A was amplified using primer 5'- GAACCCACTGCTTAAGCCTCATGGAGCCAGTAGATCCTAG and 5'-
GGGGTACCAGGTCCAGGGTTCTCCTCCACGTCTCCAGCCTGCTTCAGCAGGCTGAAGTT AGTAGCTCCGCTTCCTTCCTTCGGGCCTGTCGGGTC, respectively. These PCR products were mixed and used as a template to amplified TAR-Tat-P2A using primer 5'-
CCTCACTAAAGGTGTACAGTACTTCAAGAACTGCTGATATC and 5'-
GGGGTACCAGGTCCAGGGTTCTCCTCCACGTCTCCAGCCTGCTTCAGCAGGCTGAAGTT AGTAGCTCCGCTTCCTTCCTTCGGGCCTGTCGGGTC. A digested fragment with Bsp1407 I and Acc65 I from TAR-Tat-P2A was cloned into pPC-IP with Acc65 I to create pPCTA-IP. HA-TetOn3G was PCR amplified from pRetroX-TetOne vector (Clontech) using primer 5'-
GGGGTACCATGTACCCATACGATGTTCCAGATTACGCTTCAAGACTGGACAAG AGCAAA
G and 5'-GGCAACGCCCATCAATTGTTACCCGGGGAGCATGTCAAG. HA-TetOn3G digested with Acc65 I and Mun I was cloned into pPCTA-IP digested with Acc65 I and EcoR I to create pPCTA-TetOn3G-IP. The TRE3GS promoter was PCR amplified from pRetroX-Te-tOne vector using primers 5'-ACATGCATGCATGCATGTGGAATTATCACCTCGAG and 5'-

TATTGCCGCAATTGTTACACCACATCATCTTCGGCACCTGGCATGGCAACGCCGAATTCA CGCGTGCGGCCGCTGGATCCTTTACGAGGGTAGGAAGTGG, and the PCR product was sequentially amplified by using primers 5'-
ACATGCATGCATGCATGTGGAATTATCACCTCGAG and 5'-
AATTGACGCATGTTCGAAGAACAAACGACCCAACACCGTGCGTTTTATTCTGTCTTTTTA TTGCCGCAATTGTTACAC. The second PCR product, TRE3GS-MCS-PA tag-HSV TK poly (A) signal fragment was cloned into pPCTA-TetOn3G-IP with Mph1103 I and Bsp119 I to create pPCTetOn. Full length of SLFN11 cDNA was amplified using primers 5'-
CGTAAAGGATCCAGCATGGAGGCAAATCAGTGCC and 5'-
CGAATTCACGCGTGCCTAATGGCCACCCCACGGAA, and integrated into NotI site of the pPCTetOn vector (pPCTetON-SLFN11) using Thermo GeneArt Seamless Cloning and Assembly Enzyme Mix (A14606). The products at each step were validated by sequencing.

The expression vector containing hyperactive PB transposase cDNA under CAG promoter (pCAG2-hyPB (S1 Fig)) was first made by replacement BSD in pCAG-BSD vector with AmpR to create pCAG2. The hyPB was PCR amplified from the pCMV-hyPBase vector (kindly provided by Dr. Yusa [40]) and cloned into pCAG2. Further plasmid information can be requested to the corresponding author (J.M.).

### Generation of SLFN11-overexpressing cells

The doxycycline-inducible SLFN11 expression vector (pPCTetOn-SLFN11) and the modified expression vector of hyperactive PB transposase under CAG promoter (pCAG2-hyPB) [37] was co-transfected into SU-DHL6 cells by electroporation. One week after the transfection, cells were incubated in puromycin (0.2 μg/ml) containing medium for another 2 weeks, and surviving cells were used for the assays.

### Immunofluorescence analysis

Cells were deposited onto slide glasses (Superfrost Plus Microscope Slides precleaned, Fisher Scientific, 12-550-15) by cytospin. The deposited cells were fixed with 4% paraformaldehyde for 10 min followed by permeabilization with 0.1% Triton X-100/PBS for 15 min. The cells were incubated with 5% BSA/PBST for 30 min (blocking step). After the blocking step, the cells were incubated overnight with a primary antibody of SLFN11 (1:300 dilution) in 4% BSA/PBST in a moisture chamber at 4˚C. After washing with PBST, the cells were incubated with a proper second antibody (Alexa 488 goat anti-mouse IgG Molecular Probes cat# A11001, 1:1000 dilution) in 4% BSA/PBST for 2 hours. After washing with PBST, the cells were mounted with Vectashield with DAPI (VECTOR, H-1200). Images were captured with a Zeiss LSM 900 confocal microscope.

### Statistical analysis

For correlation analysis, Pearson's correlation was used and $p < 0.01$ was considered to be significant. For qPCR and cell count for IHC, two-sided paired t-test was used and $p < 0.05$ was considered to be significant.

## Results

### Co-expression of *SLFN11* with *XBP1* and reverse expression of *SLFN11* with *PAX5*

To understand how *SLFN11* expression is regulated during B-cell development, we mined publicly available microarray gene expression data of primary B-cells derived from healthy human bone marrow and tonsil at different developmental stages [27]. We found that, among

transcriptional regulators, the expression of *SLFN11* was most positively correlated with the expression of *XBP1*, a B-cell terminal differentiation factor, while it was most negatively correlated with *PAX5*, a master regulator of B-cell development (Fig 1A left, S2 Table) [41]. Stage-wise plotting of the data revealed that *SLFN11* expression was almost reverse of *PAX5* expression throughout the developmental stages (Fig 1A right). The data arranged from premature to differentiated B-cells showed that *SLFN11* expression was relatively low in immature B-cells, naïve B-cells, and GCBs (centroblasts and centrocytes) (Fig 1B). The pattern of *SLFN11* expression was almost parallel to the expression of *PRDM1* and *XBP1*, both of which are key transcription factors for B-cell terminal differentiation (Fig 1B) [21, 42]. By contrast, other *SLFN* family members (*SLFN5*, *12*, *13* and *14*) neither showed a marked correlation with *PAX5*, *PRDM1* nor *XBP1* (S2A and S2B Fig). Thus, among *SLFNs*, *SLFN11* uniquely showed parallel expression profile compared to *PRDM1* and *XBP1* and reverse expression profile with respect to *PAX5* across B-cell development.

Next, we mined the database Cancer Cell Line Encyclopedia (CCLE) [5] to examine *SLFN11* expression levels across different histologic subtypes of B-cell-derived cancer cell lines: B-ALL (B-cell acute lymphoblastic leukemia), GCB-DLBCL (germinal center B-cell like-diffuse large B-cell lymphoma), BL (Burkitt lymphoma), B-CLL (B-cell chronic lymphocytic leukemia), ABC-DLBCL (activated B-cell like-diffuse large B-cell lymphoma) and PCM (plasma cell myeloma). According to the origins of B-cells, the subtypes can be arranged from premature to differentiated types, and can be linked to their normal counterparts (Fig 1B below). We found that GCB-DLBCL had relatively lower expression levels of *SLFN11* while B-ALL and ABC-DLBCL expressed highest levels of *SLFN11* (Fig 1C). BL and PCM showed a broad *SLFN11* expression pattern. Thus, *SLFN11* expression levels in B-ALL, GCB-DLBCL and ABC-DLBCL appear to reflect their normal counterparts. Overall, *SLFN11* expression level is differentially regulated during B-cell development.

## Suppression of SLFN11 expression in germinal center B-cells of human lymphatic tissues

As we found that *SLFN11* is low in GCBs (centroblasts and centrocytes) and high in plasmablasts (Fig 1B), we attempted to validate the findings using dual immunohistochemical staining (IHC). Co-expression of SLFN11 and B-cell specific markers at each developmental stage was examined with multiple human normal lymphatic tissues. Fig 2A (left) shows a typical localization of B-cells at each stage of differentiation in normal lymphatic tissues. To identify the stages of B-cells, we used CD20 as a marker for premature B-cells before differentiation to plasmablasts, CD38 for plasmablasts and plasmacytes, and CD138 for plasmacytes (Fig 2A right) [43]. In GCs of tonsil tissue, CD20-positive cells were mostly negative for SLFN11, while the majority of CD38-positive cells were positive for SLFN11 (Fig 2B). CD138-positive cells were rarely found in the GCs (Fig 2B). When we focused on the outside of GCs, the mantle zone was rich of CD20-positive cells that were mostly negative for SLFN11 (Fig 2C). The cortex zone was rich of CD38-positive or CD138-positive cells that were mostly positive for SLFN11 (Fig 2C). We also performed dual IHC with two samples of spleen (S3A and S3B Fig), another tonsil sample (S4A Fig) and two samples of lymph node (S4B and S5A Figs). Similar results to Fig 2B and 2C were obtained. Additionally, we stained CD3, a general T-cell maker, and found that CD3-positive cells were mostly negative for SLFN11 (Fig 2B–2D, S3–S5 Figs). Statistically, CD38-positive or CD138-positive cells expressed SLFN11 significantly higher than CD20-positive cells (Fig 2D), which is consistent with our findings from the database analyses (Fig 1B). Collectively, our analyses reveal that SLFN11 expression changes along with B-cell development and is notably suppressed in GCBs.

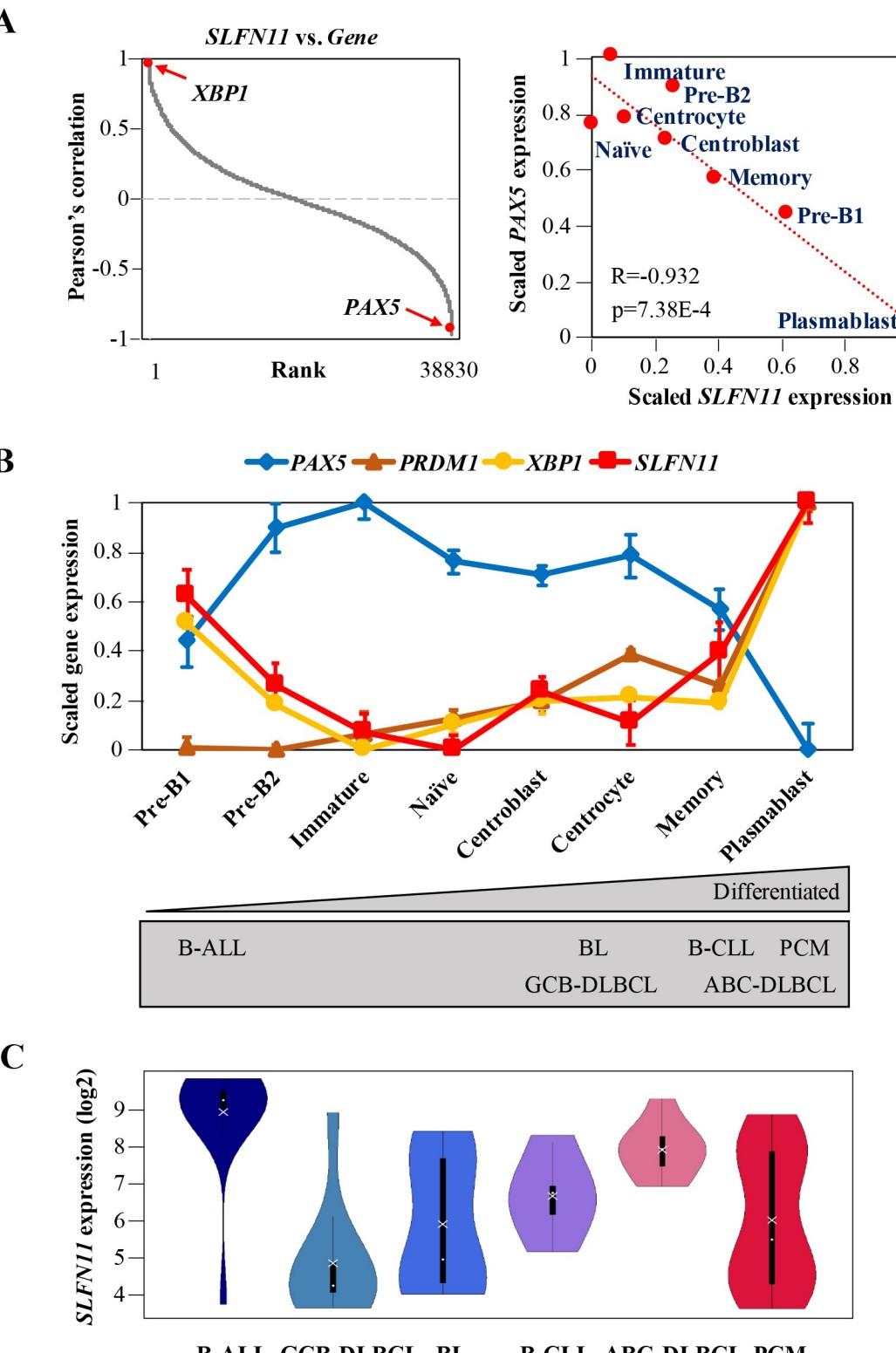

**Fig 1. SLFN11 expression is differentially regulated during B-cell development.** (A) Left: Pearson's correlation between *SLFN11* and all the other genes. The genes are ordered from the highest correlation (left) to the lowest correlation (right). Right: microarray gene expression plot of *SLFN11* and *PAX5*. Precursor (Pre)-B1 cells, precursor (Pre)-B2 and immature B-cells are were taken from human bone marrow (n = 5), and naïve B-cells, centroblasts, centrocytes, memory B-cells and plasmablasts were taken from human tonsil (n = 6). Pearson's correlation (R), P-value (p) and regression line (red dotted

line) are shown. (B) Microarray gene expression profile (log2) of selected genes (*PAX5*, *PRDM1*, *XBP1*, *SLFN11*) in human B-cells from bone marrow and tonsil. Dots correspond to group means ± SE (standard error). The gray rectangle below represents the developmental stages of the origins of B-cell-derived cancers. (C) mRNA expression (log2) of *SLFN11* in B-cell-derived cancer cell lines (B-ALL, GCB-DLBCL, BL, B-CLL, ABC-DLBCL, PCM) from CCLE. Each dot and cross correspond to the group median and mean, respectively. B-ALL, B-cell acute lymphoblastic leukemia (n = 15); GCB-DLBCL, germinal center B-cell like-diffuse large B cell lymphoma (n = 10); BL, Burkitt lymphoma (n = 11); B-CLL, B-cell chronic lymphocytic leukemia (n = 5); ABC-DLBCL, activated B-cell like-diffuse large B-cell lymphoma (n = 8); PCM, plasma cell myeloma (n = 30).

## Differential SLFN11 expression between GCB-DLBCL and ABC-DLBCL in clinical samples

DLBCL (diffuse large B-cell lymphoma) is classified into two main subtypes, GCB-DLBCL and ABC-DLBCL. In the clinic, high expression of BCL6, a transcriptional repressor required for GC formation, is used as a diagnosis indicator for GCB-DLBCL [44]. Because we have found the differential expression of *SLFN11* between GCB-DLBCL and ABC-DLBCL, we compared ABC-DLBCL and GCB-DLBCL with respect to *SLFN11* and *BCL6* expression. The correlation analysis of *SLFN11* and *BCL6* clearly separated ABC-DLBCL from GCB-DLBCL (Fig 3A).

We then investigated the expression of *SLFN11* in DLBCL tissues from patients using the database established by Reddy et al. [45]. In addition to *BCL6*, a set of genes has been reported to be distinctively expressed in the DLBCL subgroups and used to classify DLBCL [45, 46]. The set of genes includes eleven ABC-DLBCL-associated genes and eight GCB-DLBCL-associated genes (Fig 3B). Correlation analyses revealed that the expression of *SLFN11* is significantly negatively correlated to five GCB-DLBCL-associated genes (*MME*, *LRMP*, *MYBL1*, *ITPKB*, *BCL6*), whereas it is significantly positively correlated to six ABC-DLBCL-associated genes (*IRF4*, *PIMI*, *CCND2*, *ENTPD1*, *PTPN1*, *ETV6*) (Fig 3B and 3C). These results consolidate the finding of differential expression of *SLFN11* between ABC-DLBCL and GCB-DLBCL in clinical samples.

## SLFN11 expression is epigenetically suppressed in GCB-DLBCL

As epigenetic modifications are known to regulate GCB-specific genes [47–49], we hypothesized that the expression of *SLFN11* might also be epigenetically downregulated in GCBs. We first examined the correlation between *SLFN11* expression level and DNA methylation level of the *SLFN11* promoter in the dataset used in Fig 1C (Fig 4A). DNA methylation data were available in 63 out of the 79 cell lines. Overall, we found a significantly negative correlation between *SLFN11* expression and promoter DNA methylation levels (Fig 4A).

Notably, eight of the nine GCB-DLBCL cell lines had low *SLFN11* expression without promoter DNA methylation. This led us to focus on histone modification as regulator of SLFN11 expression. We tested two epigenetic modifiers, the EZH2 inhibitor tazemetostat (EPZ6438) and the histone deacetylase (HDAC) inhibitor panobinostat (LBH589), both of which have been reported to upregulate the expression of genes that are specifically suppressed in GCBs [50, 51].

We evaluated the effect of these epigenetic modifiers on the expression levels of selected GCB- and ABC-DLBCL-associated genes across the six GCB-derived lymphomas: GCB-DLBCL cell line SU-DHL6, BL cell line Daudi, and follicular lymphoma cell lines FL18, FL218 and FL318. By quantitative RT-PCR, we found that both of the epigenetic modifiers upregulated the ABC-DLBCL-associated genes, whereas they downregulated the GCB-DLBCL-associated genes in the GCB-derived lymphomas (Fig 4B). Under the same

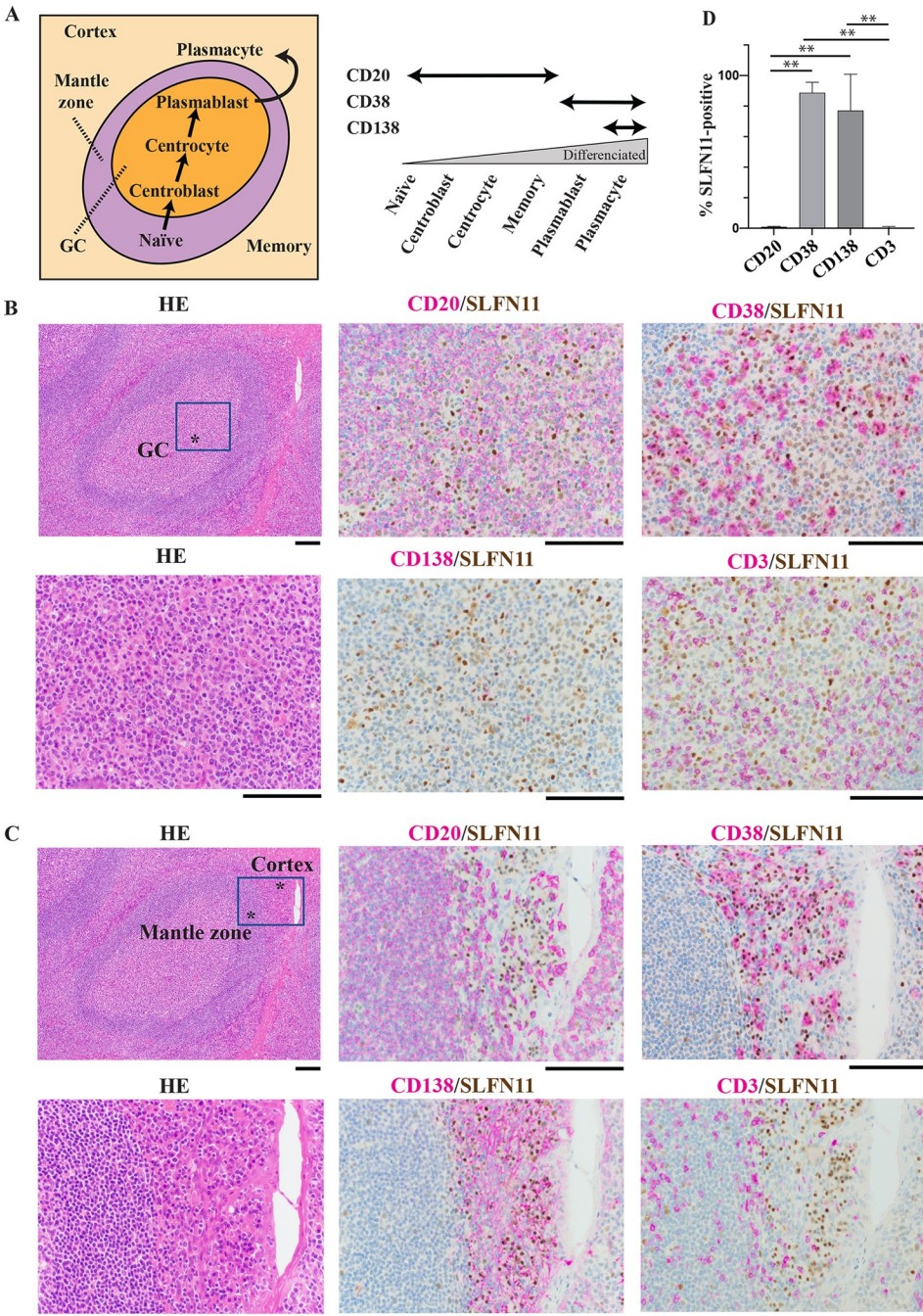

**Fig 2. SLFN11 expression is suppressed in germinal center B-cells.** Immunohistochemical staining of human tonsil tissue. The samples were stained with hematoxylin eosin (HE). For dual staining, SLFN11 was stained with DAB (brown) and CD markers (CD3, CD20, CD38 and CD138) were with HRP (purple). Original magnification: x10 and x40. Scale bars are 100 μm. GC: germinal center. (A) Left: schematic illustration of B-cell localization in lymphatic tissues. Right: CD markers expression during B-cell development. (B) Germinal center in tonsil tissue. (C) Mantle zone and cortex in tonsil tissue. (D) Proportion of SLFN11 positive cells in CD marker positive cells. **p < 0.0001.

conditions, *SLFN11* gene expression was upregulated by both of the epigenetic modifiers across all the cell lines examined (Fig 4B). The activation of SLFN11 expression was validated at the protein level in FL18 and FL318 cells after the treatment with either epigenetic modifier

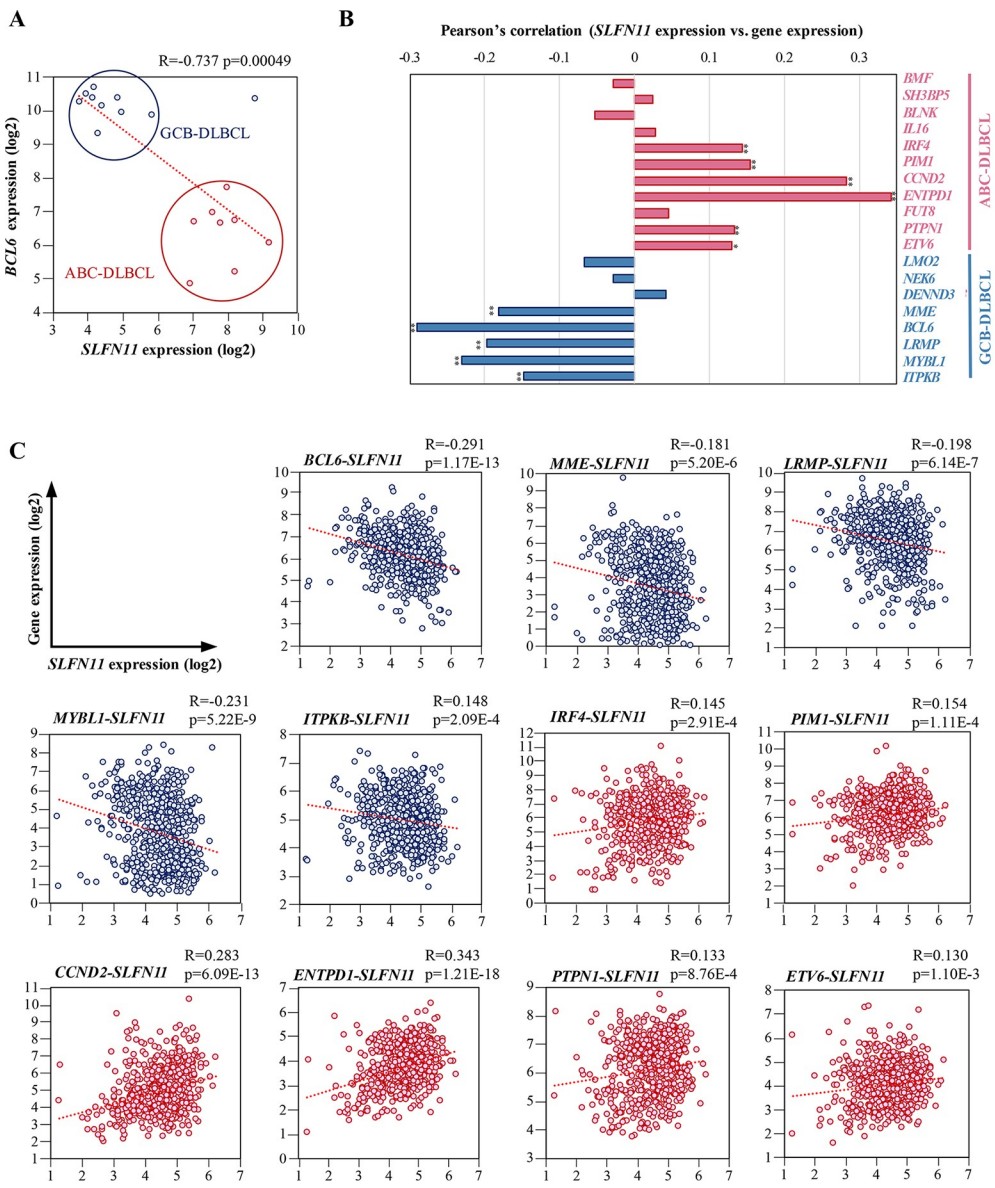

**Fig 3. Correlation of the expression of *SLFN11* and ABC- and GCB-DLBCL-associated genes in human DLBCL.**
(A) mRNA expression of *BCL6* and *SLFN11*. X-axis represents *SLFN11* mRNA expression (log2). Y-axis represents *BCL6* mRNA expression (log2). The red dots are ABC-DLBCL cell lines and the blue dots are GCB-DLBCL cell lines. Pearson's correlation (R), P-value (p) and regression line (red dotted line) are shown. GCB-DLBCL, germinal center B-cell like-diffuse large B cell lymphoma; ABC-DLBCL, activated B-cell like-diffuse large B-cell lymphoma. (B) Pearson's correlation of *SLFN11* expression level with DLBCL subtypes genes. ABC-DLBCL-associated genes consist of 11 genes (*BMF, SH3BP5, BLNK, IL16, IRF4, PIM1, CCND2, ENTPD1, FUT8, PTPN1, ETV6*). GCB-DLBCL-associated genes consist of 8 genes (*LMO2, NEK6, DENND3, MME, BCL6, LRMP, MYBL1, ITPKB*). *p < 0.01, **p < 0.001. (C) mRNA expression of DLBCL subtype genes and *SLFN11*. X-axis represents *SLFN11* mRNA expression (log2). Y-axis represents mRNA expression (log2) for genes associated with DLBCL subtypes. Pearson's correlation (R), P-value (p) and regression line (red dotted line) are shown.

([Fig 4C]). We failed to detect SLFN11 at protein levels in the other cell lines possibly because the expression levels were too low to be detected. Based on these results, we conclude that SLFN11 expression is suppressed epigenetically by histone post-translational modifications concomitantly with ABC-DLBCL-associated genes in GCB-derived lymphomas.

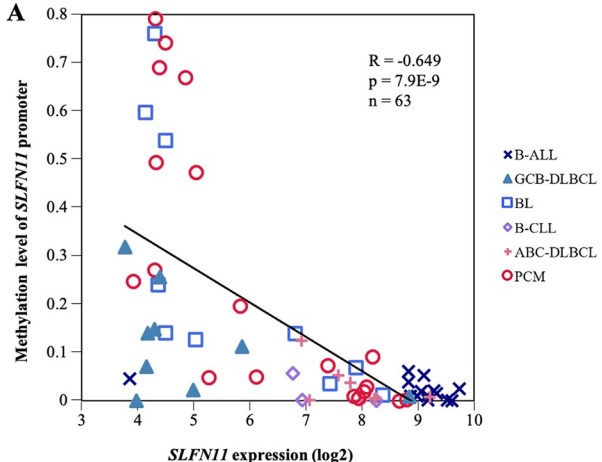

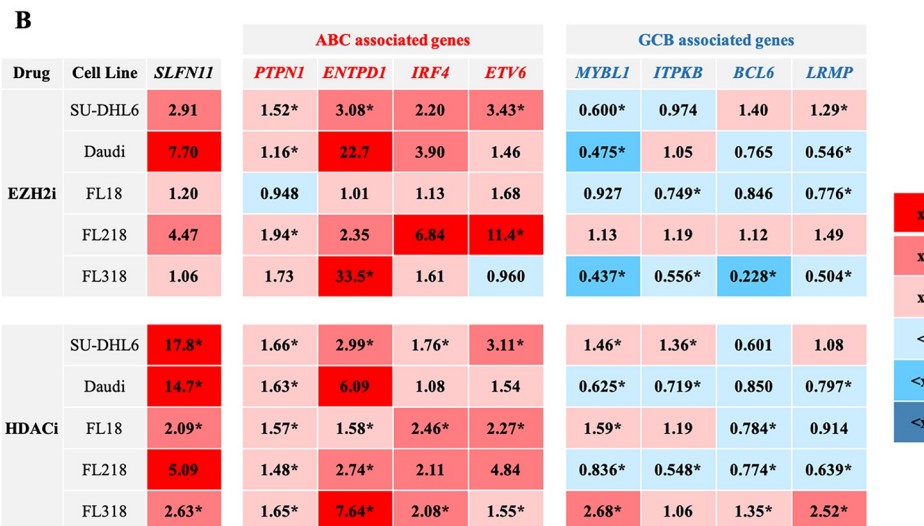

| Drug | Cell Line | *SLFN11* | ABC associated genes | | | | GCB associated genes | | | |
|---|---|---|---|---|---|---|---|---|---|---|
| | | | *PTPN1* | *ENTPD1* | *IRF4* | *ETV6* | *MYBL1* | *ITPKB* | *BCL6* | *LRMP* |
| EZH2i | SU-DHL6 | 2.91 | 1.52* | 3.08* | 2.20 | 3.43* | 0.600* | 0.974 | 1.40 | 1.29* |
| | Daudi | 7.70 | 1.16* | 22.7 | 3.90 | 1.46 | 0.475* | 1.05 | 0.765 | 0.546* |
| | FL18 | 1.20 | 0.948 | 1.01 | 1.13 | 1.68 | 0.927 | 0.749* | 0.846 | 0.776* |
| | FL218 | 4.47 | 1.94* | 2.35 | 6.84 | 11.4* | 1.13 | 1.19 | 1.12 | 1.49 |
| | FL318 | 1.06 | 1.73 | 33.5* | 1.61 | 0.960 | 0.437* | 0.556* | 0.228* | 0.504* |
| HDACi | SU-DHL6 | 17.8* | 1.66* | 2.99* | 1.76* | 3.11* | 1.46* | 1.36* | 0.601 | 1.08 |
| | Daudi | 14.7* | 1.63* | 6.09 | 1.08 | 1.54 | 0.625* | 0.719* | 0.850 | 0.797* |
| | FL18 | 2.09* | 1.57* | 1.58* | 2.46* | 2.27* | 1.59* | 1.19 | 0.784* | 0.914 |
| | FL218 | 5.09 | 1.48* | 2.74* | 2.11 | 4.84 | 0.836* | 0.548* | 0.774* | 0.639* |
| | FL318 | 2.63* | 1.65* | 7.64* | 2.08* | 1.55* | 2.68* | 1.06 | 1.35* | 2.52* |

Legend:
x5<
x2<
x1<
<x1
<x0.5
<x0.2

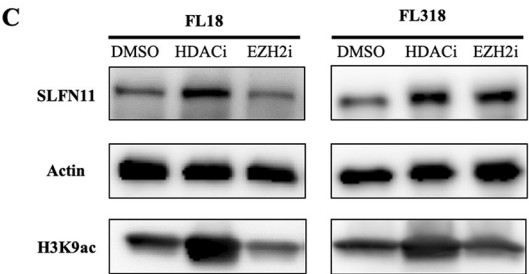

**Fig 4. Tazemetostat and panobinostat upregulate the expressions of *SLFN11* and ABC-DLBCL-associated genes in GCB-derived lymphomas.** (A) DNA methylation level of *SLFN11* promoter in B-cell-derived cancer cell lines. X-axis represents *SLFN11* mRNA expression (log2) and Y-axis represents DNA methylation level of the *SLFN11* promoter. B-ALL, B-cell acute lymphoblastic leukemia (n = 14); GCB-DLBCL, germinal center B-cell like-diffuse large B-cell lymphoma (n = 9); BL, Burkitt lymphoma (n = 10); B-CLL, B-cell chronic lymphocytic leukemia (n = 3); ABC-DLBCL, activated B-cell like-diffuse large B-cell lymphoma (n = 8); PCM, plasma cell myeloma (n = 19). Pearson's correlation (R), P-value (p), number of the samples (n) and regression line (black line) are shown. (B) Heatmap of fold changes (treated/untreated) of ABC- and GCB-DLBCL-associated genes. Cells were treated with tazemetostat (5 μM, 4 days) or panobinostat (10 nM, 16 hours) and the gene expression levels were measured by quantitative RT-PCR. Results are the average of three independent experiments. *p < 0.05 (two-sided paired t-test). (C) Western blotting of SLFN11, H3K9ac and Actin in FL18 and FL318 treated with tazemetostat (5 μM, 16 hours) or panobinostat (10 nM, 16 hours).

## Epigenetic upregulation of SLFN11 renders GCBs more susceptible to cytosine arabinoside

Cytosine arabinoside (AraC, cytarabine), a replication inhibitor, is one of the therapeutic options for patients with B-cell-derived cancers. Data of drug activity (inhibitory concentration 50%: IC50) and RNA-seq data of ~1000 human cancer cell lines are available from the Genomics of Drug Sensitivity in Cancer (GDSC; https://www.cancerrxgene.org) and the NCI CellMiner databases (https://discover.nci.nih.gov/cellminercdb). Drug activity data of AraC and *SLFN11* expression data were accessible in 39 cell lines out of the 79 B-cell-derived cancer cell lines used in Fig 1C. Correlation analysis reveals that *SLFN11* expression is significantly correlated with the activity of AraC (Fig 5A left), indicating the potential utility of SLFN11 expression as predictor of drug activity for AraC in B-cell-derived cancers. In addition to AraC, the activity of camptothecin (CPT), a topoisomerase inhibitor was also found correlated to *SLFN11* expression in 31 B-cell-derived cancer cell lines available in the GDSC database (Fig 5A right).

Next, we examined the potential synergistic effect of AraC on SU-DHL6 cells with or without pretreatment of the EZH2 inhibitor (tazemetostat) or the HDAC inhibitor (panobinostat) to induce SLFN11 expression. We found that the pretreatment with these epigenetic modifiers enhanced cell susceptibility to AraC in SU-DHL6 cells (Fig 5B and 5C left). Combination index (CI) value was used to evaluate synergistic effects of the combination, and revealed that concurrent treatment of AraC with tazemetostat or panobinostat exhibited synergistic effects at various doses (Fig 5B and 5C right).

To test whether SLFN11 enhances the susceptibility to DNA damaging agents, we generated doxycycline-inducible SLFN11-overexpressing SU-DHL6 (SU-DHL6 tetON SLFN11) cells. The overexpression of SLFN11 was confirmed by western blot and immunofluorescence (Fig 5D). SLFN11 overexpression made SU-DHL6 cells more susceptible to AraC and CPT (Fig 5E). These results indicate that the induction of SLFN11 can improve the therapeutic response to AraC in GCB-derived lymphomas.

## Discussion

In this study, we show that SLFN11 expression is differentially regulated during B-cell development. We find that SLFN11 is typically suppressed in GCBs (centroblasts and centrocytes) and GCB-derived lymphomas. The suppression is partly achieved epigenetically, and is reversible with the EZH2 inhibitor tazemetostat or the HDAC inhibitor panobinostat, which increase the cytocidal function of AraC. These results suggest that these combinations could be applied to the treatment of GCB-derived lymphomas having low SLFN11.

Physiological reasons why SLFN11 needs to be suppressed in GCBs are not biologically examined in this study. However, in GCBs, activation-induced cytidine deaminase (AID) is specifically highly expressed to introduce somatic hypermutations in variable regions of the immunoglobulin genes, while AID can also induce DNA damage at non-target genes by generating apurinic sites [52]. Furthermore, GCBs proliferate very rapidly [50], which can result in replication-dependent DNA damage [53]. Hence, GCBs are at high risk of eliciting a DNA damage response due to AID expression and rapid proliferation. Because SLFN11 exclusively executes replicating cells carrying genotoxic stress [7], we speculate that SLFN11 needs to be downregulated in GCBs to avoid SLFN11-dependent cell death in response to physiological genomic rearrangements in GCBs.

We then questioned which gene(s) are associated with *SLFN11* expression in GCBs. We found that almost perfect inverse correlation between *SLFN11* expression and *PAX5*, a B-cell lineage-specific repressor (Fig 1A and 1B). By mining a public database of chromatin

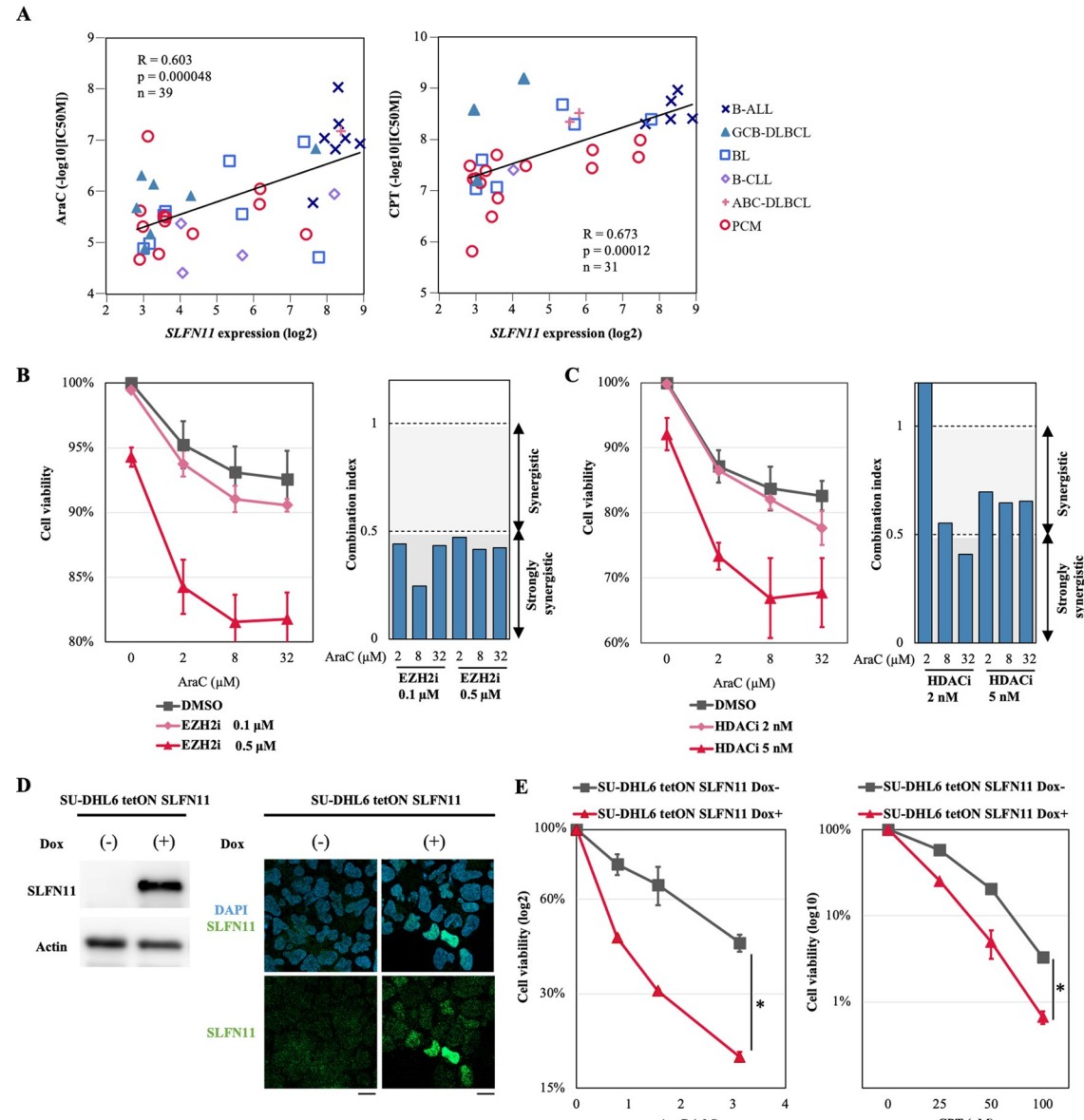

**Fig 5. Combination of cytosine arabinoside (AraC) with epigenetic modifiers is synergistic in SU-DHL6 cells.** (A) Efficacy of cytosine arabinoside (AraC) and camptothecin (CPT) for B-cell-derived cancer cell lines is correlated with *SLFN11* expression. X-axis represents *SLFN11* mRNA expression (log2) and Y-axis represents IC50 for AraC or CPT (log10). B-ALL, B-cell acute lymphoblastic leukemia (n = 7 for AraC, n = 5 for CPT); GCB-DLBCL, germinal center B-cell like-diffuse large B-cell lymphoma (n = 7 for AraC, n = 3 for CPT); BL, Burkitt lymphoma (n = 8 for AraC, n = 6 for CPT); B-CLL, B-cell chronic lymphocytic leukemia (n = 4 for AraC, n = 1 for CPT); ABC-DLBCL, activated B-cell like-diffuse large B-cell lymphoma (n = 1 for AraC, n = 2 for CPT); PCM, plasma cell myeloma (n = 12 for AraC, n = 14 for CPT). Pearson's correlation (R), P-value (p), number of the samples (n) and regression line (black line) are shown. (B) and (C) Left: Cell viability evaluated by propidium iodide staining. SU-DHL6 cells were pretreated with tazemetostat (0.1 μM or 0.5 μM) for 4 days or panobinostat (2 nM or 5 nM) for 16 hours, and then treated with the indicated concentrations of AraC (X-axis) for another 24 hours before cell viability was assessed by propidium iodide staining and flow cytometry. Cell viability (%) was defined as treated cells/untreated cells × 100 (Y-axis). Results are the average of three independent experiments with ± SD (standard deviation). Right: Combination index (CI) value assessed using the CompuSyn Software for data points of tazemetostat or panobinostat in combination with AraC. Shading represents the levels of synergism (> 1; synergy, > 0.5; strong synergy). (D) SLFN11 expression in SU-DHL6 tetON SLFN11 cells treated with or without doxycycline (1 μg/mL) for 48 hours was measured by western blotting (left) and immunofluorescence (right). Scale bars are 10 μm. (E) Cell viability of SU-DHL6 tetON SLFN11 cells treated by AraC or CPT. SU-DHL6 tetON SLFN11 cells were pretreated with doxycycline (1 μg/mL) for 24 hours, and then treated with the indicated concentrations of AraC or CPT (X-axis) for another 96 hours before cell viability was assessed. The ATP level in untreated cells was defined as 100%. Cell viability (%) was defined as ATP level of treated cells/ATP level of untreated cells × 100. Results are representative of three independent experiments with ± SD (standard deviation). *p < 0.05 (two-sided paired t-test).

immunoprecipitation-sequencing for PAX5 [54], we found a potential PAX5 binding site (GCGTGAC) in the promoter region of *SLFN11*, suggesting that PAX5 may be one of the repressors of *SLFN11* in B-cells. This possibility is also supported by the fact that *SLFN11* expression is parallel to the expression of *PRDM1* and *XBP1*, which are the targets of PAX5 (Fig 1B).

Epigenetic regulation of *SLFN11* has been reported in other malignancies. In small cell lung cancer cells, *SLFN11* expression is silenced by marked deposition of H3K27me3, leading to drug resistance, and is reactivated by inhibition of EZH2 a methyltransferase for H3K27 [24]. The EZH2 inhibitor, tazemetostat has recently been approved by the FDA for the treatment of follicular lymphoma and its efficacy for DLBCL is being studied [55]. In leukemia K562 and fibrosarcoma HT1080 cell lines, both of which have a very low basal SLFN11 expression, HDAC inhibitors (romidepsin and entinostat) increase SLFN11 expression and enhance sensitivity to DNA-damaging agents in SLFN11-dependent manner [26]. Our data consolidate these findings with GCB-derived lymphoma cell lines and provide a rationale to treat B-cell lymphoma with low SLFN11 expression by combining tazemetostat with AraC. Moreover, this is the first report showing that SLFN11 can be physiologically regulated through histone modifications during normal developmental process.

As SLFN11 is a promising target to sensitize tumor cells to cytotoxic chemotherapy, regulatory factors of SLFN11 expression are also favorable targets for cancer treatment [56]. Our findings of dynamic regulation of SLFN11 during B-cell development will provide a basis to further investigate potential regulatory factors of SLFN11 at different developmental stages including PAX5 and histone modifiers.

## Supporting information

**S1 Fig. Plasmids for generation of SLFN11-overexpressing cells.**
(TIF)

**S2 Fig. *SLFN* family members are differently expressed during B-cell development.** (A) Upper: Pearson's correlation between *SLFN* family members *(SLFN5*, *SLFN12*, *SLFN13*, *SLFN14)* and all the other genes. The genes are ordered from the highest correlation (left) to the lowest correlation (right). Lower: microarray gene expression plot of *SLFN* family members and *PAX5*. Precursor (Pre)-B1 cells, precursor (Pre)-B2 and immature B-cells are were taken from human bone marrow (n = 5), and naïve B-cells, centroblasts, centrocytes, memory B-cells and plasmablasts were taken from human tonsil (n = 6). Pearson's correlation (R), P-value (p) and regression line (red dotted line) are shown. (B) Microarray gene expression profile (log2) of selected genes (*PAX5*, *PRDM1*, *XBP1*, *SLFN* family members) in human B-cells from bone marrow and tonsil. Dots correspond to group means ± SE.
(TIF)

**S3 Fig. SLFN11 expression is suppressed in germinal center B-cells in spleen tissue.** Immunohistochemical staining of human spleen tissue. The samples were stained with hematoxylin eosin (HE). For dual staining, SLFN11 was stained with DAB (brown) and CD markers (CD3, CD20, CD38 and CD138) were with HRP (purple). Original magnification: x10 and x40. Scale bars are 100 μm. GC: germinal center. (A) Germinal center in spleen tissue. (B) Mantle zone and cortex in spleen tissue.
(TIF)

**S4 Fig. SLFN11 expression is suppressed in germinal center B-cells in tonsil and lymph node tissue.** Immunohistochemical staining of human tonsil and lymph node tissue. The samples were stained with hematoxylin eosin (HE). For dual staining, SLFN11 was stained with

DAB (brown) and CD markers (CD3, CD20, CD38 and CD138) were with HRP (purple). Original magnification: x10 and x40. Scale bars are 100 μm. GC: germinal center. (A) Germinal center, mantle zone and cortex in tonsil tissue. (B) Germinal center in lymph node tissue. (TIF)

**S5 Fig. SLFN11 expression is upregulated in plasmablasts and plasmacytes in lymph node tissue.** Immunohistochemical staining of human lymph node tissue. The samples were stained with hematoxylin eosin (HE). For dual staining, SLFN11 was stained with DAB (brown) and CD markers (CD3, CD20, CD38 and CD138) were with HRP (purple). Original magnification: x10 and x40. Scale bars are 100 μm. (A) Non-germinal center region in lymph node tissue. (TIF)

**S1 Table. Primer sequences used for RT-qPCR.**
(XLSX)

**S2 Table. Pearson's correlation with *SLFN11* expression during B-cell development.**
(XLSX)

## Acknowledgments

We would like to thank Drs. A. Reddy and S. S. Dave (Duke University) for kindly providing access to the DLBCL database, Ms. F. Sasaki for technical assistance and Dr. M. Tomita for general support (Institute for Advanced Biosciences, Keio University) and Dr. H. Saya for kindly providing the vector for SLFN11 overexpression (Institute for Advanced Medical Research, Keio University School of Medicine). This work was partly conducted through the Joint Research Program of the Radiation Biology Center, Kyoto University.

## Author Contributions

**Conceptualization:** Fumiya Moribe, Momoko Nishikori, Junko Murai.

**Formal analysis:** Fumiya Moribe, Hiroshi Arima.

**Funding acquisition:** Momoko Nishikori, Hiroyuki Sasanuma, Shunichi Takeda, Yves Pommier, Akifumi Takaori-Kondo, Junko Murai.

**Investigation:** Fumiya Moribe, Tsuyoshi Takashima, Daiki Taniyama, Nobuyuki Onishi, Remi Akagawa, Fathi Elloumi, Junko Murai.

**Methodology:** Fumiya Moribe, Momoko Nishikori, Hiroyuki Sasanuma, Yves Pommier, Junko Murai.

**Supervision:** Momoko Nishikori, Shunichi Takeda, Eiichi Morii, Akifumi Takaori-Kondo, Junko Murai.

**Writing – original draft:** Fumiya Moribe, Momoko Nishikori, Junko Murai.

**Writing – review & editing:** Fumiya Moribe, Yves Pommier, Junko Murai.

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
