## [Decision Letter · Decision Letter 0]

14 Aug 2020

PONE-D-20-23146

Epigenetic suppression of SLFN11 in germinal center B cells in the process of the dynamic expression change during B-cell development

PLOS ONE

Dear Dr. Mural,

Thank you for submitting your manuscript to PLOS ONE. After careful consideration, we feel that it has merit but does not fully meet PLOS ONE’s publication criteria as it currently stands. Therefore, we invite you to submit a revised version of the manuscript that addresses the points raised during the review process.

Specifically, both reviewers pointed out that the conclusions were not solidly supported by the data and the data were over-interpreted in multiple places. 

We look forward to receiving your revised manuscript.

Kind regards,

Wei Xu

Academic Editor

PLOS ONE

Journal Requirements:

2.PLOS ONE now requires that authors provide the original uncropped and unadjusted images underlying all blot or gel results reported in a submission’s figures or Supporting Information files. This policy and the journal’s other requirements for blot/gel reporting and figure preparation are described in detail at https://journals.plos.org/plosone/s/figures#loc-blot-and-gel-reporting-requirements and https://journals.plos.org/plosone/s/figures#loc-preparing-figures-from-image-files. When you submit your revised manuscript, please ensure that your figures adhere fully to these guidelines and provide the original underlying images for all blot or gel data reported in your submission. See the following link for instructions on providing the original image data: https://journals.plos.org/plosone/s/figures#loc-original-images-for-blots-and-gels.

3.Thank you for stating the following in the Competing Interests section:

[I have read the journal's policy and the authors of this manuscript have the following competing interests:

M.N. and A.T-K received honorarium and research funding from Eisai Co., Ltd. Other authors declare no conflicts of interest.].

Reviewers' comments:

Reviewer's Responses to Questions

**Comments to the Author**

1. Is the manuscript technically sound, and do the data support the conclusions?

Reviewer #1: Partly

Reviewer #2: Partly

2. Has the statistical analysis been performed appropriately and rigorously? 

Reviewer #1: N/A

Reviewer #2: Yes

3. Have the authors made all data underlying the findings in their manuscript fully available?

Reviewer #1: No

Reviewer #2: Yes

4. Is the manuscript presented in an intelligible fashion and written in standard English?

Reviewer #1: No

Reviewer #2: Yes

5. Review Comments to the Author

Reviewer #1: SLFN11, a putative DNA/RNA helicase, has been implicated in DNA damage response to platinum-derivatives, topoisomerase inhibitors, PARP inhibitors and replication inhibitors. SLFN11 expression has been shown regulated by epigenetic modifications of DNA and histones in different cell types and tissues. In this paper, Fumiya Moribe et. al. investigated the regulatory mechanisms of SLFN11 expression and its potential roles in B cells by mining publicly available microarray gene expression data and experimentally tested the relationship between SLFN11 expression using IHC and cell viability. They found that SLFN11 expression is epigenetically regulated during B-cell differentiation, and it is typically suppressed in germinal center B cells(GCBs). Moreover, epigenetic activation of SLFN11 in lymphomas of GCB origin enhanced the susceptibility of lymphoma cells to a DNA-damaging agent.

Overall the experimental design is logical and the data analysis method is sound. A major weakness is that not all the data are supportive of their conclusions and some results appear to be preliminary and not validated experimentally. There are many grammar and spelling errors; the manuscript should be proofread by a native English speaker. In addition, the manuscript should be carefully revised to clarify the rationales and the results need to be interpreted properly. The specific comments are listed below:

1. Line 182-183, the results do not support this conclusion;

2. Figure 2: It needs to clarify how many samples were used in the IHC experiments. A statistical analysis should be performed with clearly labeled sample numbers. In addition, SLFN11 is also not expressed in B cells in Non-GCB regions. How SLFN11 expression is regulated in B cells in non-GCB regions should be explained.

3. The dynamic expression of SLFN11 during B cell development should be validated in both mouse or human samples using IHC.

4. Line 305-307, the results do not reconcile with this conclusion. To draw the conclusion, the authors should examine whether SLFN11 depletion can sensitize the cells to cytosine arabinoside(AraC).

5. Line 355-356, the results do not support this conclusion.

6. Figure 4. B the quality of Western blot result is poor. The bands of Actin and H3K9ac in FL318 cells are vague. RT-PCR results need to be statistically analyzed.

Reviewer #2: In this manuscript, the authors gave us a general investigation that SLFN11 was downregulated in germinal center B cells, and epigenetic modifiers EZH2 inhibitor and HDAC inhibitor could elevate its expression, which drove the cells more sensitive to cytosine arabinoside treatment.

Comments:

1. in the text, some of the conclusions were over interpreted, like line 182-183, line 262.

2. No any evidence to show the mechanism or SLFN11 downregulation;

3. Knockdown or overexpression of SLFN11 gene in ABC or GCB cell lines to study the sensitivity of these cells to arabinoside treatment.

6. PLOS authors have the option to publish the peer review history of their article (what does this mean?). If published, this will include your full peer review and any attached files.

Reviewer #1: No

Reviewer #2: No

---

## [Author Response · Author response to Decision Letter 0]

11 Nov 2020

Point-by-point answers to the reviewers 

The list of changes and answers are indicated below with the comments of the reviewers included before our answers to facilitate the review process.

Reviewers' comments to the Author

Reviewer #1: SLFN11, a putative DNA/RNA helicase, has been implicated in DNA damage response to platinum-derivatives, topoisomerase inhibitors, PARP inhibitors and replication inhibitors. SLFN11 expression has been shown regulated by epigenetic modifications of DNA and histones in different cell types and tissues. In this paper, Fumiya Moribe et. al. investigated the regulatory mechanisms of SLFN11 expression and its potential roles in B cells by mining publicly available microarray gene expression data and experimentally tested the relationship between SLFN11 expression using IHC and cell viability. They found that SLFN11 expression is epigenetically regulated during B-cell differentiation, and it is typically suppressed in germinal center B cells(GCBs). Moreover, epigenetic activation of SLFN11 in lymphomas of GCB origin enhanced the susceptibility of lymphoma cells to a DNA-damaging agent.

Overall the experimental design is logical and the data analysis method is sound. A major weakness is that not all the data are supportive of their conclusions and some results appear to be preliminary and not validated experimentally. There are many grammar and spelling errors; the manuscript should be proofread by a native English speaker. In addition, the manuscript should be carefully revised to clarify the rationales and the results need to be interpreted properly. 

(Answer) Thank you very much for your positive comments and constructive suggestions. In the revised manuscript, we added new data to consolidate our conclusions. They are included in the revised Figure 2 & Figures S2-S4 (dual immunohistochemical staining for SLFN11 and CD20/CD38/CD138/CD3), Figure 4A (DNA methylation plot for SLFN11 promoter) and Figure 5A (drug activity data for AraC across 39 B-cell-derived cancers). We have also carefully checked and edited our manuscript to clarify the rationale of our experiments and the interpretation of the results. Moreover, the revised manuscript has been proofread by a native English speaker. Thank you.

The specific comments are listed below:

1. Line 182-183, the results do not support this conclusion;

(Answer) Thank you for your suggestions. In the revised manuscript, we write “Thus, among SLFNs, SLFN11 uniquely showed parallel expression profile compared to PRDM1 and XBP1 and reverse expression profile with respect to PAX5 across B-cell development.” instead of the original sentence “Thus, SLFN11 but no other SLFNs expression can be controlled during B-cell development under the same regulatory system for PRDM1 and XBP1.” 

2. Figure 2: It needs to clarify how many samples were used in the IHC experiments. A statistical analysis should be performed with clearly labeled sample numbers. In addition, SLFN11 is also not expressed in B cells in Non-GCB regions. How SLFN11 expression is regulated in B cells in non-GCB regions should be explained.

(Answer) Thank you very much for your suggestions. As suggested, in the revised manuscript, we increased the number of samples as well as B-cell markers to consolidate our conclusions. We employed six samples from three lymphatic tissue types (including each two lymph nodes, tonsils and spleen samples). In addition to the dual staining of CD20 (a marker of premature B cells) and SLFN11, we performed dual staining of CD38 and CD138 (markers of differentiated B-cells) with SLFN11 to determine SLFN11 expression in plasmablasts and plasmacytes. Overall, the results revealed that SLFN11 is downregulated in GCBs (CD20-positive cells in GCs) and is upregulated in plasmablasts (CD38-positive in GCs) and plasmacytes (CD38-positive and CD138-positive in cortex). We added these results in Figure 2A-C and Figure S2-4 in the revised manuscript. Moreover, we scored the SLFN11-positive population (%) to statistically support our findings by visual inspection. While only 0.8% of CD20-positive cells were SLFN11-positive, 89% and 77% of CD38-positive and CD138-positive cells, respectively, were SLFN11-positive (Figure 2D, **p<0.0001). 

Additionally, to distinguish T-cells from B-cells, we performed dual staining for CD3 (a marker of T cells) and SLFN11. We found that 0.5% of CD3-positive cells were SLFN11-positive (Fig2B-D).

As for the upregulation of SLFN11 in non-GCB regions, we could not clarify the precise mechanism of the upregulation in this study. However, our data (Figs 3-5) imply that histone modification could be key for the regulation of SLFN11 in B-cells. This point has been included in our revised manuscript.

3. The dynamic expression of SLFN11 during B cell development should be validated in both mouse or human samples using IHC.

(Answer) Thank you. To validate the dynamic expression change of SLFN11 during B-cell development, we performed dual IHC with several markers for B-cells (Fig 2). Please see our answers to point # 2 above. Importantly, validation in mouse samples is currently not feasible as SLFN11 ortholog has not been identified among the different SLFN11 murine genes.

4. Line 305-307, the results do not reconcile with this conclusion. To draw the conclusion, the authors should examine whether SLFN11 depletion can sensitize the cells to cytosine arabinoside(AraC).

(Answer) Thank you for your suggestions. To answer your question, we worked hard to make SLFN11 KO cells in SU-DHL6 and other GCB cell lines. However, SU-DHL6 did not express SLFN11 high enough to be detected by WB under the treatments of epigenetic modifiers, and we were unable to validate candidate clones of SLFN11-KO. For follicular lymphoma cell lines, FL18, FL218 and FL318 did not form single colonies after the transfection of SLFN11-KO vectors. For these reasons, we have been unable to establish SLFN11-KO cell lines in the GCB cells used in this study. Alternatively, to examine the relationship between SLFN11 expression and sensitivity to AraC in GCB cells, we mined publicly available data on SLFN11 expression and drug activity of AraC. SLFN11 expression was found significantly correlated to the activity of AraC in B-cell-derived cancer cell lines. These data have been included in the revised Figure 5A. Moreover, we previously reported that the synergistic effect by the combination of HDACi and a DNA damaging agent (camptothecin) is SLFN11-dependent (i.e., the synergy was not observed in SLFN11-KO cells) (Tang SW, et al. Clin Cancer Res. 2018. [26]). Based on this, we conclude that activation of SLFN11 expression by epigenetic modifiers can enhance the activity of AraC. Nevertheless, since we failed to generate KO cells in GCB cells, we rephrased the original sentence and stated, “These results indicate that the combination of the epigenetic modifiers that enhance SLFN11 expression can improve the response in GCB lymphoma cells to AraC” in the revised manuscript.

5. Line 355-356, the results do not support this conclusion.

(Answer) An increased number of studies indicate that SLFN11 is a plausible target to sensitize tumor cells to cytotoxic chemotherapy (Coussy F, et al. Sci Transl Med. 2020 [17], Gardner EE, et al. Cancer Cell. 2017 [24], Nogales V, et al. Oncotarget. 2016 [25] and more). Hence, factors that regulate SLFN11 expression are legitimate therapeutic targets for further testing (Murai J, et al. Pharmacol Ther. 2019 [4] and Tang SW, et al. Clin Cancer Res. 2015 [52]). Therefore, at the end of the discussion of our revised manuscript, we would like to retain this point to foster future studies by independent investigators.

6. Figure 4. B the quality of Western blot result is poor. The bands of Actin and H3K9ac in FL318 cells are vague. RT-PCR results need to be statistically analyzed.

(Answer) Thank you for carefully examining our results. As suggested, we performed new Western blots and used better images in the revised manuscript (Fig 4C). Regarding RT-PCR, we performed t-test for all the genes in the heat map (Fig 4B) and added asterisks to represent the significant changes. We also omitted the bar-graphs for fold change of SLFN11 in Figure 4B right of the original manuscript to avoid showing redundant data.

Reviewer #2: In this manuscript, the authors gave us a general investigation that SLFN11 was downregulated in germinal center B cells, and epigenetic modifiers EZH2 inhibitor and HDAC inhibitor could elevate its expression, which drove the cells more sensitive to cytosine arabinoside treatment.

Comments:

1. in the text, some of the conclusions were over interpreted, like line 182-183, line 262.

(Answer) Thank you very much for these suggestions. Regarding the conclusions in lines 182-183, we revised the sentence to “Thus, among SLFNs, SLFN11 uniquely showed parallel expression profile compared to PRDM1 and XBP1 and reverse expression profile with respect to PAX5 across B-cell development.” For line 262, to avoid overinterpretation we revised the sentence to “These results consolidate the finding of differential expression of SLFN11 between ABC-DLBCL and GCB-DLBCL in clinical samples.”

2. No any evidence to show the mechanism of SLFN11 downregulation;

(Answer) Thank you for raising an important point. To determine whether epigenetics is a mechanism for SLFN11 downregulation, we examined the correlation between SLFN11 expression and DNA methylation level of the SLFN11 promoter in B-cell-derived malignancies. Overall, we observed a significant reverse correlation between SLFN11 expression and DNA methylation across B-cell derived cancers (revised Figure 4A). Interestingly, SLFN11 expression of GCB-DLBCL lines (▲) is suppressed without promoter DNA methylation, implying that histone modifications rather than DNA methylation are plausible suppressors of SLFN11 suppression in GCBs. Consistent with this possibility, we found that HDAC inhibitor or EZH2 inhibitor reactivate SLFN11 expression in GCB cells (revised Figure 4B-C).

3. Knockdown or overexpression of SLFN11 gene in ABC or GCB cell lines to study the sensitivity of these cells to arabinoside treatment.

(Answer) 

Thank you for your suggestions. To answer your question, we worked hard to make SLFN11 KO cells in SU-DHL6 and other GCB cell lines. However, SU-DHL6 did not express SLFN11 high enough to be detected by WB under the treatments of epigenetic modifiers, and we were unable to validate candidate clones of SLFN11-KO. For follicular lymphoma cell lines, FL18, FL218 and FL318 did not form single colonies after the transfection of SLFN11-KO vectors. For these reasons, we have been unable to establish SLFN11-KO cell lines in the GCB cells used in this study. Alternatively, to relate SLFN11 expression with the activity of AraC in GCB cells, we mined publicly available data on SLFN11 expression and drug activity of AraC. SLFN11 expression was found significantly correlated to the activity of AraC in B-cell-derived cancer cell lines. These data have been included in the revised Figure 5A. This result is consistent with a previous report showing that the synergistic combination of HDACi with the replication selective DNA damaging agents (camptothecin) is SLFN11-dependent (i.e., the synergy was not observed in SLFN11-KO cells) (Tang SW, et al. Clin Cancer Res. 2018. [26]). Based on these observations, we consider that activation of SLFN11 expression by epigenetic modifiers can enhance the activity of AraC. Nevertheless, since we failed to generate KO or OE cells in GCB cells, we rephrased our conclusion in the revised manuscript and stated, “These results indicate that the combination of the epigenetic modifiers that enhance SLFN11 expression can improve the response in GCB lymphoma cells to AraC.”

---

## [Decision Letter · Decision Letter 1]

23 Nov 2020

PONE-D-20-23146R1

Epigenetic suppression of SLFN11 in germinal center B-cells during B-cell development

PLOS ONE

Dear Dr. Murai,

Thank you for submitting your manuscript to PLOS ONE. After careful consideration, we feel that it has merit but does not fully meet PLOS ONE’s publication criteria as it currently stands. Therefore, we invite you to submit a revised version of the manuscript that addresses the points raised during the review process.

Reviewer one requested repeating of WB of SLFN11 and performed overexpression or knockdown of SLFN11.

We look forward to receiving your revised manuscript.

Kind regards,

Wei Xu

Academic Editor

PLOS ONE

Reviewers' comments:

Reviewer's Responses to Questions

**Comments to the Author**

1. If the authors have adequately addressed your comments raised in a previous round of review and you feel that this manuscript is now acceptable for publication, you may indicate that here to bypass the “Comments to the Author” section, enter your conflict of interest statement in the “Confidential to Editor” section, and submit your "Accept" recommendation.

Reviewer #1: (No Response)

Reviewer #2: All comments have been addressed

2. Is the manuscript technically sound, and do the data support the conclusions?

Reviewer #1: Partly

Reviewer #2: Yes

3. Has the statistical analysis been performed appropriately and rigorously? 

Reviewer #1: Yes

Reviewer #2: Yes

4. Have the authors made all data underlying the findings in their manuscript fully available?

Reviewer #1: Yes

Reviewer #2: Yes

5. Is the manuscript presented in an intelligible fashion and written in standard English?

Reviewer #1: Yes

Reviewer #2: Yes

6. Review Comments to the Author

Reviewer #1: SLFN11 plays key role in execute cancer cells harboring replicative stress induced by DNA damaging agents while the roles of SLFN11 under physiological conditions are not widely studied. B-cells undergo gene editing at variable regions of the immunoglobulin gene loci during the development and maturation. During this process, B-cells are physiologically exposed to genotoxic stress caused by somatic hypermutations and class-switch recombination. Such genotoxic tress is introduced particularly to centroblasts and centrocytes in germinal centers (GCs) of lymph nodes. Thus, Germinal center B-cells (GCBs) undergo somatic hypermutations and class-switch recombination, which can cause physiological genotoxic stress. Hence, Fumiya Moribe etal tested whether the expression of SLFN11 is needed to be controlled during B-cell development to avoid SLFN11- dependent cell death in cells undergoing genomic rearrangements. They performed several mRNA and Protein expression analysis on SLFN11 by using cell lines of different stages of normal B cells and various types of B-cell lymphoma or normal human lymphatic tissues and some Cell viability experiments and found the following results: SLFN11 mRNA level was low in both normal GCBs and GCB-DLBCL (GCB like-diffuse large 4 B-cell lymphoma). Low SLFN11 expression in GCBs and high SLFN11 expression in plasmablasts and plasmacytes. The EZH2 and HDAC epigenetic modifiers upregulated SLFN11 expression in GCB-derived lymphomas and made them more susceptible to cytosine arabinoside. Overall the experimental designs are logical and some data mincing are meaningful. The major concern is that the article is focus on the expression and function of SLFN11 during B cell development, especially in the GCB-derived lymphomas and their susceptibility to cytosine arabinoside. Therefore, SLFN11 knock down or overexpression in some of these cells are necessary for further studying. A few minor comments are listed below:

1. Figure 4C, the western blot bands are not clear. Especially H3K9AC. The right SLFN11 band should be divided with the non specific bands by extending the running time of SDS page gels.

2. Figure 5, SLFN11 should be knockdown by shRNA (SiRNA) or overexpressed in ABC or GCB cell lines to study the sensitivity of these cells to arabinoside treatment.

Reviewer #2: (No Response)

7. PLOS authors have the option to publish the peer review history of their article (what does this mean?). If published, this will include your full peer review and any attached files.

Reviewer #1: No

Reviewer #2: No

---

## [Author Response · Author response to Decision Letter 1]

5 Dec 2020

Point-by-point answers to the reviewers 

The list of changes and answers are indicated below with the comments of the reviewers included before our answers to facilitate the review process.

Reviewers' comments to the Author

Reviewer #1: SLFN11 plays key role in execute cancer cells harboring replicative stress induced by DNA damaging agents while the roles of SLFN11 under physiological conditions are not widely studied. B-cells undergo gene editing at variable regions of the immunoglobulin gene loci during the development and maturation. During this process, B-cells are physiologically exposed to genotoxic stress caused by somatic hypermutations and class-switch recombination. Such genotoxic tress is introduced particularly to centroblasts and centrocytes in germinal centers (GCs) of lymph nodes. Thus, Germinal center B-cells (GCBs) undergo somatic hypermutations and class-switch recombination, which can cause physiological genotoxic stress. Hence, Fumiya Moribe etal tested whether the expression of SLFN11 is needed to be controlled during B-cell development to avoid SLFN11- dependent cell death in cells undergoing genomic rearrangements. They performed several mRNA and Protein expression analysis on SLFN11 by using cell lines of different stages of normal B cells and various types of B-cell lymphoma or normal human lymphatic tissues and some Cell viability experiments and found the following results: SLFN11 mRNA level was low in both normal GCBs and GCB-DLBCL (GCB like-diffuse large 4 B-cell lymphoma). Low SLFN11 expression in GCBs and high SLFN11 expression in plasmablasts and plasmacytes. The EZH2 and HDAC epigenetic modifiers upregulated SLFN11 expression in GCB-derived lymphomas and made them more susceptible to cytosine arabinoside. Overall the experimental designs are logical and some data mincing are meaningful. The major concern is that the article is focus on the expression and function of SLFN11 during B cell development, especially in the GCB-derived lymphomas and their susceptibility to cytosine arabinoside. Therefore, SLFN11 knock down or overexpression in some of these cells are necessary for further studying. A few minor comments are listed below:

1. Figure 4C, the western blot bands are not clear. Especially H3K9AC. The right SLFN11 band should be divided with the non specific bands by extending the running time of SDS page gels.

(Answer) Thank you for carefully examining our results. As suggested, we repeated the western blot for FL218 for multiple times. However, due to the very low expression of SLFN11 in FL218, we were unable to divide the SLFN11 band from the non-specific band. Thus, we tried to test SLFN11 protein level in another cell line. We could observe SLFN11 bands in FL18 and its upregulation by HDACi, which is consistent with the qPCR results. Hence, we included the western blot results of FL18 and excluded those of FL218 in the updated manuscript. As for the blot of H3K9Ac, we confirmed that HDACi but not EZH2 inhibitor increased the acetylation level of H3K9, which is reasonable since HDACi targets broad acetylation sites of histones while EZH2 inhibitor targets exclusively H3K27. Hence, we keep the previous blot of H3K9Ac in FL318 and added a comparable result of FL18.

2. Figure 5, SLFN11 should be knockdown by shRNA (SiRNA) or overexpressed in ABC or GCB cell lines to study the sensitivity of these cells to arabinoside treatment.

(Answer) Thank you very much for your suggestions. We generated SLFN11-overexpressing SU-DHL6 cell line by using piggy back tetON system (SU-DHL6 tetON SLFN11 cells). In this system, we were able to induce SLFN11 expression when treating the cells with doxycycline. SLFN11 induction in SU-DHL6 tetON SLFN11 cells was confirmed by western blot and immunofluorescence (Fig5D). Using this cell line, we examined whether SLFN11 sensitizes the cells to DNA damaging agents. We treated SU-DHL6 tetON SLFN11 cells with arabinoside (AraC) and observed that SU-DHL6 tetON SLFN11 cells were more sensitive to AraC (Fig 5E left). 

We also found SLFN11 expression was correlated to the activity of camptothecin (CPT) in B-cell cancer cell lines in the GDSC database (Fig5A right). SLFN11 overexpression sensitized the cells to CPT as well (Fig5E right). To consolidate the role of SLFN11 in cellular sensitivity to DNA damaging agents in GCBs, we included these data in the revised manuscript.

---

## [Editor Report · Decision Letter 2]

8 Dec 2020

Epigenetic suppression of SLFN11 in germinal center B-cells during B-cell development

PONE-D-20-23146R2

Dear Dr. Mural,

We’re pleased to inform you that your manuscript has been judged scientifically suitable for publication and will be formally accepted for publication once it meets all outstanding technical requirements.

Kind regards,

Wei Xu

Academic Editor

PLOS ONE

---

## [Editor Report · Acceptance letter]

20 Jan 2021

PONE-D-20-23146R2 

Epigenetic suppression of SLFN11 in germinal center B-cells during B-cell development 

Dear Dr. Murai:

I'm pleased to inform you that your manuscript has been deemed suitable for publication in PLOS ONE. Congratulations! Your manuscript is now with our production department. 

Kind regards, 

on behalf of

Dr. Wei Xu 

Academic Editor

PLOS ONE